# Hierarchical Dialogue Understanding with Special Tokens and Turn-level Attention

**Xiao Liu, Jian Zhang,**\* **Heng Zhang,**\* **Fuzhao Xue, Yang You**
Department of Computer Science, National University of Singapore
{liuxiao, zhang.jian, zhang_heng, f.xue}@u.nus.edu,
youy@comp.nus.edu.sg

## Abstract

Compared with standard text, understanding dialogue is more challenging for machines as the dynamic and unexpected semantic changes in each turn. To model such inconsistent semantics, we propose a *simple but effective* **Hi**erarchical **Dialog**ue Understanding model, HiDialog. Specifically, we first insert multiple special tokens into a dialogue and propose the turn-level attention to learn turn embeddings hierarchically. Then, a heterogeneous graph module is leveraged to polish the learned embeddings. We evaluate our model on various dialogue understanding tasks including dialogue relation extraction, dialogue emotion recognition, and dialogue act classification. Results show that our simple approach achieves state-of-the-art performance on all three tasks above. All our source code is publicly available at https://github.com/ShawX825/HiDialog.

## 1 Introduction

Task-oriented dialogue system (TODS) plays a key role in assisting users to complete tasks automatically, and a well-trained model in TODS can help to save money and time for people. Different from the formal text, dialogues (e.g., meetings, interviews, and debates) deliver intertwined, inconsistent semantic information, where each turn forms a unit of information gain, fulfilling the needs of participating interlocutors. This unfavorable dynamic is caused by speaker intent disparity, conversation progression, and abrupt change of thought. Such dynamics in dialogue are usually ignored by large pre-trained language models. For BERT-style pre-trained models (Devlin et al., 2018), $[CLS]$ token is applied to model the sentence-level semantics during pre-training. However, dialogue-level natural language understanding requires methods to capture both intra-turn and inter-turn information. Although there are existing works using the special tokens method to enhance dialogue understanding, most of them involve an additional pre-training stage (Shen et al., 2021; Li et al., 2021; Chapuis et al., 2020). Given that the cost of such a pre-training stage grows exponentially with model size, it is unlikely that school labs would have the resources necessary to pre-train such models on sizable dialogue datasets. Thus, we devote ourselves to bridging the gap between BERT-style pre-training and dialogue understanding fine-tuning, without extra computational cost and training data.

## 2 Method

**Problem Formulation**. The dialogue understanding task aims at building a model $f$ that can make a prediction on a query text given a dialogue. Specifically, the input includes a multi-turn dialogue ($\{s_i : t_i | i \in [1, m]\}$), and a query text $q$ with $k$ arguments ($q_1, q_2, ..., q_k$), where $i, s_i, t_i$ denotes $i^{th}$ turn, the corresponding speaker ID and text. The output is predicted class $\hat{y}$ of the query text $q$.

**Input Module**. For a given multi-turn dialogue $d$ and a query $q$, we follow Yu et al. (2020) to reconstruct $d$ as $\hat{d} = \{\zeta(s_i) : t_i | i \in [0, m]\}$, and the arguments in query $q$ as $\hat{q}_j = \zeta(a_j)$, where $\zeta(\cdot)$ maps token $x_i$ and argument $q_j$ to a special token $[S_j]$ when $x_i = q_j$. Then, we concatenate reconstructed dialogue $\hat{d}$ and query $\hat{q}$ with a special token $[CLS]$ and separator tokens $[SEP]$. To leverage speaker information, we add speaker embedding (Gu et al., 2020) into our input sequence.

**Intra-turn Modeling**. Prior approaches have leveraged either a global $[CLS]$ token to capture sentence-level semantics (Devlin et al., 2018) or have initialized turn embeddings by simply averaging over tokens (Lee & Choi, 2021). However, we argue that these methods have not sufficiently emphasized the significance of certain tokens that carry crucial information (e.g., trigger words in

---

\*Equal contributions

DialogRE (Yu et al., 2020)), which may result in less discriminative learned turn embeddings. To capture intra-turn information, we insert a special token $\tau_i$ ahead of each turn $t_i$ and arguments, where a weighted sum of token embeddings can be learned by the self-attention mechanism in the encoder. Moreover, turn-level attention is proposed to avoid these special tokens functioning as standard special tokens. Specifically, tokens not belonging to a certain turn $t_i$ are masked out for their corresponding turn-level special token $\tau_i$. This approach compels each turn-level special token to act as an information aggregator of its own turn. Note that a global $[CLS]$ token is placed ahead of the whole sequence, these two types of special tokens then form a hierarchical way to gather information in the encoder module. The intra-turn modeling is illustrated in Appendix A.1.

**Inter-turn Modeling**. To model the interaction between turns and entities, we establish a heterogeneous graph $\mathcal{G} = (\mathcal{V}, \mathcal{E})$ from the output of the encoder. $\mathcal{G}$ contains three types of nodes: dialogue node, turn node, and argument node. The embedding of each node is initialized from the corresponding special token, i.e., global classification token $h_{[CLS]}$ for dialogue node, turn-level classification tokens $h_\tau$ for turn nodes, and argument nodes. We follow Lee & Choi (2021) to establish different types of edges: dialogue edge, speaker edge, and entity edge, sequence edge. See Appendix A.2 for more details on graph edges. To enable nodes to gather information via different types of edges, we apply Graph Transformer Network (Yun et al., 2019) to further polish the turn embeddings.

**Classification**. Dialogue and argument nodes obtained from $\mathcal{G}$ are concatenated and fed into a linear classifier to generate the final prediction. Cross entropy loss is used as the object function.

## 3 EXPERIMENTS

**Overall Performance**. We first evaluated HiDialog on the Dialogue Relation Extract (DRE) dataset, DialogRE (Yu et al., 2020) and the Emotion Recognition in Conversation (ERC) dataset, MELD (Poria et al., 2019). Further details on datasets and metrics can be found in Appendix A.3. We selected BERT (Devlin et al., 2018), GDPNet (Xue et al., 2021), RoBERTa$_s$ (Yu et al., 2020), SimpleRE (Xue et al., 2022), and TUCORE-GCN (Lee & Choi, 2021) as baselines. As reported in Table 1, HiDialog established new state-of-the-art results on both datasets.

| Method | DialogRE | | MELD |
|---|---|---|---|
| | $F1$ | $F1_c$ | $F1$ |
| BERT | 57.9 | 53.1 | 61.31 |
| GDPNet | 60.2 | 57.3 | - |
| RoBERTa$_s$ | 71.3 | 63.7 | 64.19 |
| SimpleRE | 66.7 | - | - |
| TUCORE-GCN$^\dagger$ | 73.1 | 65.9 | 65.36 |
| HiDialog$^\dagger$ | **77.1**$_{+4.0}$ | **68.2**$_{+2.3}$ | **66.96**$_{+1.6}$ |

Table 1: All methods performance on the DialogRE and MELD, averaged over five runs. $^\dagger$ uses RoBERTa as the encoder.

**Towards Generality**. Our intra-turn modeling's simplicity suggests its potential as a valuable solution for enhancing dialogue understanding without the need for extra pre-training. To assess this claim, we integrated it into the baseline encoder without any additional components, such as an inter-turn module or speaker embeddings. For fair comparisons, only the encoder's global $[CLS]$ token was used in a softmax classifier for prediction.

| Method | MELD | ENLP | DDialog | MRDA | DialogRE | |
|---|---|---|---|---|---|---|
| | | | | | $F1$ | $F1_c$ |
| PHT | 61.90 | - | 60.14 | **92.4** | - | - |
| DialogXL | 62.41 | 34.73 | 54.93 | - | - | - |
| RoBERTa$_s$ | 64.19 | 38.03 | 61.65 | 91.3 | 71.3 | 63.7 |
| +Intra-turn | **65.64**$_{+1.45}$ | **38.13**$_{+0.1}$ | **61.83**$_{+0.28}$ | 91.5$_{+0.2}$ | **74.4**$_{+3.1}$ | **66.6**$_{+2.9}$ |

Table 2: All methods performance on 5 multi-turn dialogue-based understanding datasets: MELD, EmoryNLP, DailyDialog, MRDA, DialogRE, averaged over five runs. Performance gains over the RoBERTa$_s$ are highlighted in green.

We conducted the experiment on 5 datasets from 3 different tasks: DRE (DialogRE), ERC (MELD, EmoryNLP (Zahiri & Choi, 2018), DailyDialog (Li et al., 2017)), and Dialogue Act Classification (MRDA (Shriberg et al., 2004)). We chose RoBERTa$_s$, Pretrained Hierarchical Transformer (PHT) (Chapuis et al., 2020), and DialogXL (Shen et al., 2021) as baselines. Compared to PHT and DialogXL, both of which require additional pre-training to address the domain adaption gap, the performance of proposed intra-turn modeling is surprisingly good in all 5 datasets (Table 2). Moreover, we conducted an ablation study and analysis, which further reveals HiDialog is good at handling asymmetric relations and robust against increasing utterance length (see Appendix A.4).

In conclusion, our HiDialog provides a *simple yet effective* approach to fill the gap between general corpus pre-training and dialogue understanding, without extra computational cost and training data while maintaining decent performance. Therefore, we anticipate that it could serve as a compelling baseline or plug-in module for future work in the community.

## CONTRIBUTION

This work has been expanded upon from a course project that was undertaken by the key authors, Xiao Liu, Jian Zhang, and Heng Zhang, during their time as students at the National University of Singapore. Fuzhao Xue, the teaching assistant, and project mentor played a role in guiding and shaping the outcome of this work. We are grateful to Yang You for providing us with valuable instructions and computational resources.

## ACKNOWLEDGEMENT

Yang You's research group is being sponsored by NUS startup grant (Presidential Young Professorship), Singapore MOE Tier-1 grant, ByteDance grant, ARCTIC grant, SMI grant and Alibaba grant. Our sincere appreciation also goes out to Min-Yen Kan, our course lecturer, for his invaluable suggestions during our enrollment.

## URM STATEMENT

The authors acknowledge that the first author of this work meets the URM criteria of ICLR 2023 Tiny Papers Track.

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

## A    APPENDIX

### A.1    ILLUSTRATION OF INTRA-TURN MODULE

### A.2    EDGES TYPES IN INTER-TURN MODULE

We follow Lee & Choi (2021) to establish four different types of edges in graph $\mathcal{G}$: Dialogue edge, to learn global information across the whole dialogue, all turn nodes are connected to the dialogue node; Speaker edge, every pair of turn nodes belongs to the same speaker are connected; Entity edge, an argument node is connected to a turn node if it is mentioned in this turn. To enable the model to directly learn the sequence information, every pair of turn nodes in a graph is connected via sequence edge.

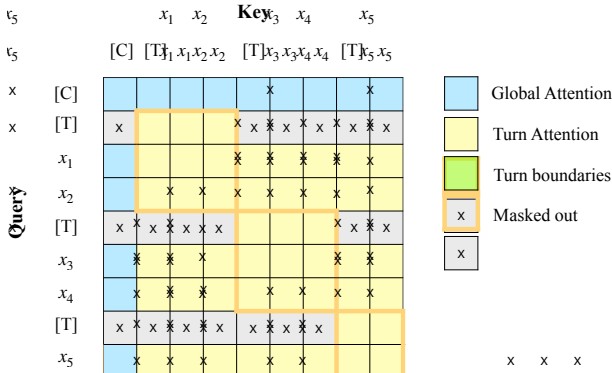

Figure 1: Illustration of the proposed intra-turn modeling. In the turn-level attention, the restriction is applied on turn-level special tokens, denoted as $[T]$, where tokens outside the turn are masked out (colored in grey).

## A.3    DATASETS AND METRICS

**DialogRE** (Yu et al., 2020) is a relation extraction task based on 1,788 dialogues from the Friends transcript. Each pair of arguments can be classified as one of 36 possible relation types. For each of the 10,168 human-annotated entity pairs, the trigger words are also provided.

**EmoryNLP** (Zahiri & Choi, 2018) is an emotion detection task based on 12,606 utterances from the Friends transcript. Each utterance can be classified as one of seven emotions, e.g., joyful, scared.

**DailyDialog** (Li et al., 2017) is a dialogue database containing 13,118 simple English dialogues. Each utterance can be assigned an emotion label from seven categories (anger, surprise, etc.).

**MELD** (Poria et al., 2019) is an emotion detection task based on 13,000 sentences from the Friends transcript. Each utterance can be classified as one of eight emotions, such as sad, disgust.

**MRDA** (Shriberg et al., 2004) is a dialogue act task based on 75 hours of real-life meeting transcript. Each sentence is assigned a general dialogue act (topic change, repeat, etc.) and a specific dialogue act (apology, suggestion, etc.).

**Metrics**. For DialogRE, F1 and $F1_c$ are used as evaluation metrics. $F1_c$ modifies F1 by taking an early part of the dialogue as input Yu et al. (2020). For MELD and EmoryNLP, we use weighted-F1 as metrics. For DailyDialog, the Micro-F1 score excluding the neutral class is used as the metric.

## A.4    ABLATION STUDY AND ANALYSIS

**Ablation study on components.** We conduct an ablation study on DialogRE to evaluate the effectiveness of critical components in our method: turn-level attention, turn-level special tokens, and graph module. The results are reported in Table 3. First, we remove the turn-level attention mask. It can be observed that performance slightly drops. The inserted turn-level special tokens function as normal separators after the mask has been removed. Thus, these special tokens are able to aggregate information from the entire sequence, thus they are

| Method | F1 | F1$_c$ |
|--------|-----|--------|
| HiDialog | 77.1 | 68.2 |
| w/o attention mask | 76.5 (-0.6) | 67.9 (-0.3) |
| w/o special tokens | 75.6 (-1.5) | 67.4 (-0.8) |
| only intra-turn | 74.4 (-2.7) | 66.6 (-1.6) |

Table 3: Ablation Study on HiDialog components on DialogRE to evaluate the individual effect of turn-level attention, turn-level special tokens, and graph module.

not context-aware at the turn level. Second, we remove the turn-level attention along with turn-level special tokens. The only difference between our final model and the ablated one is in how we initialize the node embedding of the graph module, that is, in this experiment, we average corresponding tokens for the initialization. The $F1$ score decreases by 1.5% and the $F1_c$ score declines by 0.8%, which indicates the turn-level attention is effective.

**Analysis of relations**. We grouped the test set of DialogRE according to the relation types into three subsets: (I) asymmetric, when a relation type differs from its inversion (e.g. *children* and *parents*); (II) symmetric, when a relation type is the same as its inversion (e.g. *spouse*); (III) other, when a relation type does not have inversion (e.g. *age*). We compared the performance of our model with baselines and report the results in Table 4. As we can observe, there is a great performance increase in the asymmetric subset while the F1 score drops moderately for symmetric relations. This trend reverses when we remove the graph module in our method (i.e. symmetric > asymmetric).

**Analysis of robustness against increasing utterance length.** With the hierarchical aggregation in HiDialog, each turn-level special token is enforced to capture intra-turn critical information regardless of the whole dialogue. This nature enables our method to handle dialogues of various lengths. Motivated by it, we further divide the samples in the DialogRE test set into six groups according to their lengths and report the F1 score for each group achieved by our method and previous SOTA, TUCORE-GCN. As shown in Figure 2, our method consistently outperforms TUCORE-GCN in all groups, where the largest performance gap can be found in the group with less than 100 tokens. Moreover, TUCORE-GCN shows a great drop with an increase of length (i.e., from $[400, 500)$ to $[500, +\infty)$), while HiDialog maintains decent performance for long sequences.

| Method | I | II | III |
|---|---|---|---|
| BERT | 42.5 | 60.7 | 65.6 |
| GDPNet | 47.4 | 59.8 | 68.1 |
| RoBERTa$_s$ | 57.4 | 69.3 | 79.6 |
| TUCORE-GCN | 62.3 | **71.3** | 79.9 |
| HiDialog | **76.6** | 70.5 | **80.9** |
| w/o graph module | 65.5 | 69.9 | 79.4 |

Table 4: All methods performance on DialogRE. We break down the performance into three groups (I) asymmetric inverse relations, (II) symmetric inverse relations, and (III) others.

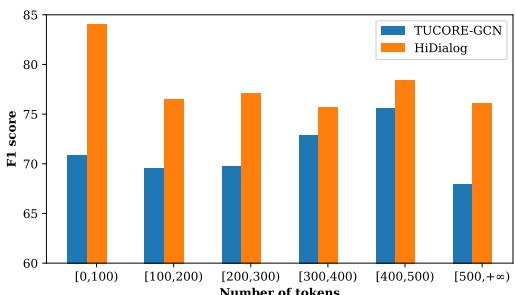

Figure 2: Analysis of robustness of HiDialog tackling increasing utterance length compared to baseline TUCORE-GCN on DialogRE dataset.

## A.5 RELATED WORK

**Multi-Turn Dialogue Understanding**. Various tasks and corresponding benchmarks are proposed to evaluate the capacities of dialogue understanding models. Dialogue-based relation extraction (RE) is a classification task that assigns a pair of entities a relation label in a dialogue. Focusing on the word level, Xue et al. (2021) constructed a multi-view graph with words in the dialogue as nodes and proposed Dynamic Time Warping Pooling to automatically select words in interest. SimpleRE (Xue et al., 2022) designed a novel input sequence format and utilized a Relation Refinement Gate to filter the semantic representation which is later fed into the classifier. TUCORE-GCN (Zahiri & Choi, 2018) used a heterogeneous dialogue graph to encode the interaction between speakers, arguments and turns across the dialogues.

Emotion Recognition in Conversation (ERC) has been extensively studied in the research community. It aims to attach an emotional label to every turn in a given dialogue. Kratzwald et al. (2018) customized the recurrent neural network with bidirectional processing to solve the problem of emotion classification. Majumder et al. (2019) leveraged the Recurrent Neural Network to extract the information of the party states and use it to predict the emotion in conversations with two speakers. On top of the recurrent neural network, COSMIC Ghosal et al. (2020) models the commonsense knowledge, mental states, events, and actions to enhance emotion detection in dialogue.

Deep learning based methods have been extensively studied in recent works (Lee & Dernoncourt, 2016; Chen et al., 2018; Raheja & Tetreault, 2019) regarding Dialogue Act classification (DAC). Chen et al. (2018) introduced a relation layer into the shared hierarchical encoder to model the interaction between the tasks of dialog act recognition and sentiment classification.

**Context-Aware Representation Learning**. To address dynamics and semantic changes in multi-turn dialogue, previous works extend pre-trained large language models to learn context-aware representations for turns (Lee & Choi, 2021; Shen et al., 2021; Li et al., 2021; Chapuis et al., 2020).

TUCORE-GCN (Lee & Choi, 2021) proposes the turn attention module, masking out distant turns to learn the contextual embeddings. Instead of adding extra modules, DialogXL (Shen et al., 2021) targets the encoder and incorporates four self-attention mechanisms to different attention heads to capture diverse dialog-aware information. Similarly, such dialogue-oriented self-attention can also be found in MDFN (Liu et al., 2021) where it is defined as utterance-aware and speaker-aware channels. However, most of them involve an additional pre-training stage (Shen et al., 2021; Li et al., 2021; Chapuis et al., 2020).

