# OpenReview forum: "Hierarchical Dialogue Understanding with Special Tokens and Turn-level Attention"
_ICLR.cc/2023/TinyPapers — Submitted to Tiny Papers @ ICLR 2023_

### Official Review · Reviewer_c3NV · 2023-03-29

**Confidence:** 4

**Summary Of Contributions:**

This paper presents a simple yet effective dialogue understanding model by modeling intra-turn and inter-turn semantics.

**Rating:**

Great Start (GS): a submission which meets some of the reviewing criteria but has room for improvement

**Strengths And Weaknesses:**

Summary:

This paper proposes a HiDialog, a Hierarchical Dialogue Understanding model. Special tokens are inserted in embedding that needs no more pre-training but leverage self-attention to represent intra-turn information. A graph is applied for aggregating inter-turn information. Experimental results on four datasets show the effectiveness of the proposed method.

Strength:

The experiments involve five datasets containing tasks of relation extraction, emotion detection, and act detection. The results show the effectiveness of HiDialog.

Weakness:

1. The innovation compared with existing methods is not clear. The embedding method is the same as Yu et al. 2020 and Gu et al.. The turn-level self-attention with mask for intra-turn modeling has been widely explored [1,2]. The graph module is also from Lee & Choi, 2021, and has been applied to many related methods in the last several years [3,4].

2. Some descriptions of Section 2 are a little ambiguous. Some expressions may need clarification, i.e., “ discriminative learned turn embeddings”,“weighted sum of token embeddings”, “avoid these special tokens functioning as standard special tokens”.

Question: What is the advantage of “only intra-turn” over Lee & Choi, 2021 since the same graph is applied?

3. The motivation for being free from pre-training is not that convincing as lots of dialogue comprehension methods require no further pre-training, only fine-tuning.

[1] Filling the Gap of Utterance-aware and Speaker-aware Representation for Multi-turn Dialogue

[2] Self- and Pseudo-self-supervised Prediction of Speaker and Key-utterance for Multi-party Dialogue Reading Comprehension

[3] Graph-Based Knowledge Integration for Question Answering over Dialogue

[4] DialogueGCN: A Graph Convolutional Neural Network for Emotion Recognition in Conversation

**Suggested Changes:**

1. A clear comparison with existing studies is needed.

2. Some expressions need clarification. Please see the detailed comments above.

---

### Official Review · Reviewer_JDTC · 2023-03-30

**Confidence:** 4

**Summary Of Contributions:**

Authors suggest a turn-level attention based method to improve dialogue understanding. The method is evaluated on multiple benchmarks.

**Rating:**

Great Start (GS): a submission which meets some of the reviewing criteria but has room for improvement

**Strengths And Weaknesses:**

Strengths

1. The paper is very detailed in terms of method description and experimental results.
2. Evaluation is performed on multiple datasets and compared with multiple baselines.

Weaknesses

1. The paper would be best presented as a Short Paper instead of a Tiny Paper. There are multiple methodologies presented in the paper and it distracts from the coherence of the paper presentation given the page limit. Many important sections such as ablation studies and related works are pushed to Appendix, meaning that the paper content itself is already at 4 pages.
2. The methodology is only described via equations instead of diagrams. For example, intra-turn modeling regarding special tokens, and inter-turn modeling regarding graph nodes, would be best presented as a diagram. The equations are also not clear and are parsed with text.
3. Probably because of the page limit, the paper does not have a discussion section or conclusions. The conclusion presented in the experiments is a bit too concise and just repeats the abstract.

**Suggested Changes:**

1. Please consider revising the paper as a short paper instead of a tiny paper.
2. Add diagrams of the methodology.
2. Add a discussion/conclusion section.

---

### Official Review · Reviewer_s6qR · 2023-03-31

**Confidence:** 3

**Summary Of Contributions:**

This paper introduces HiDialog, a simple and effective approach that incorporates turn-level attention and special turn-level tokens for dialogue understanding. The model is extensively evaluated on three different tasks, demonstrating state-of-the-art performance.

**Rating:**

Clear, Correct, and Reproducible (CCR): a submission which meets the reviewing criteria

**Strengths And Weaknesses:**

Strengths:
1. The authors present an innovative approach, incorporating turn-level attention for intra-turn modeling and a heterogeneous graph module for inter-turn modeling.
2. The comprehensive experimental evaluation of HiDialog reveals its effectiveness, achieving state-of-the-art performance across the three tasks under consideration.

Weaknesses:
1. The inclusion of qualitative examples of the model's performance on dialogue understanding tasks would help readers better understand the strengths and limitations of the proposed approach.
2. Providing figures or diagrams illustrating the model's structure would help reader understand the model's components.

**Suggested Changes:**

I would recommend that the authors include more visualizations and qualitative demonstrations.

---

### Meta-Review · Area_Chair_y3XL · 2023-04-05

**Recommendation:** Invite to archive
**Confidence:** 4

**Metareview:**

This paper introduces HiDialog, a simple and effective approach that incorporates turn-level attention and special turn-level tokens for dialogue understanding. Evaluation is performed on multiple datasets, demonstrating state-of-the-art performance. But the description of the methodology can be a little ambiguous without the support of diagrams or qualitative demonstrations. More comparisons with existing studies and more specific discussion/conclusion sections are needed.



**Summary:**

The authors present an approach for dialogue understanding, which incorporats turn-level attention for intra-turn modeling and a heterogeneous graph module for inter-turn modeling. The reviewers noted the strength of comprehensive experimental evaluations but raised concerns about clarity of methodology and experiments, as well as comparison with existing methods.

**Comments And Feedback To The Authors:**

1．The writing needs to be polished. Providing figures or diagrams would better illustrate the model's structure and improve comprehension of the model's components.

2．Please make more clarification on the innovation and advantages compared to existing methods.

3．Please consider revising the discussion and conclusion section to emphasize the advantages.

**Reason For Not Giving A Higher Recommendation:**

This paper presents an approach to dialogue understanding that has several limitations.

(i) The methodology is not well presented as equations are used instead of diagrams, making it difficult to understand certain aspects of the approach.

(ii) The paper lacks clarity in terms of the innovation compared to existing methods and the advantage of the proposed approach over previous work.

(iii) The absence of a discussion section and qualitative examples of the model's performance on dialogue understanding tasks diminishes the clarity and effectiveness of the presentation.

Overall, addressing these issues would greatly improve the paper's quality and effectiveness.

**Reason For Not Giving A Lower Recommendation:**

I would not recommend “Invite to revise” as the method proposed in this paper HiDialog, is simple and effective, and evaluated on multiple datasets, demonstrating state-of-the-art performance, which all the reviewers agree on.

---

### Decision · Program_Chairs · 2023-04-10

Invite to archive

---

> ### Author Response · Authors · 2023-05-30
> **Confirmation of archival**
>
> We confirm that we opt for the archival. We would like also to thank the reviewers for their constructive suggestions.